# Study on the Road Friction Database for Automated Driving: Fundamental Consideration of the Measuring Device for the Road Friction Database

Ichiro Kageyama [1,2,*], Yukiyo Kuriyagawa [2], Tetsunori Haraguchi [1,2,3,*], Tetsuya Kaneko [4], Motohiro Asai [5] and Gaku Matsumoto [5]

1 Consortium on Advanced Road-Friction Database, 1-4-31 Hachimandai, Sakura 285-0867, Japan
2 College of Industrial Technology, Nihon University, 1-2-1 Izumi-cho, Narashino 275-8575, Japan; kuriyagawa.yukiyo@nihon-u.ac.jp
3 Institutes of Innovation for Future Society, Nagoya University, Furocho, Chikusaku, Nagoya 464-8603, Japan
4 Department of Mechanical Engineering for Transportation, Faculty of Engineering, Osaka Sangyo University, 3-1-1 Nakagaito, Daito 574-8530, Japan; kaneko@ge.osaka-sandai.ac.jp
5 Nihon Michelin Tire Co., Ltd., Shinjuku Park Tower 13F, 3-7-1 Nishi-Shinjuku, Shinjuku-ku, Tokyo 163-1073, Japan; motohiro.asai@michelin.com (M.A.); gaku.matsumoto@michelin.com (G.M.)
* Correspondence: ichiro.kageyama@car-fd.or.jp (I.K.); haraguchi@nagoya-u.jp (T.H.)

**Featured Application: The magic formula (MF) is used to estimate the road friction characteristics.**

**Abstract:** This study deals with the possibility of construction of a database on the braking friction coefficient for actual roads from the viewpoint of traffic safety, especially for automated driving, such as level 4 or higher. At these levels of automated driving, the controller needs to control the vehicle. However, the road surface condition, especially the road friction coefficient on wet roads and snowy or icy roads, changes greatly, and in some cases, changes by almost one order. Therefore, it is necessary for the controller to constantly collect environment information, such as the road friction coefficients, and prepare for emergencies, such as obstacle avoidance. However, at present, the measurement of the road friction coefficients is not systemically performed, and a method for accurately measuring has not been established. In order to improve this situation, this study examines a method for continuous measurement of the road friction characteristics, such as the $\mu$-$s$ characteristics. It is shown that the $\mu$-$s$ characteristics are continuously measured using the MF generally used in tire engineering, and the friction characteristics identified from the results are sufficiently satisfactory.

**Keywords:** road friction; environmental information; measurement; friction estimation

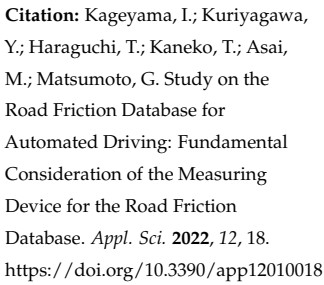



## 1. Introduction

According to data from the Ministry of Land, Infrastructure, Transport, and Tourism, the pavement rate of Japanese roads in 2019 is about 82.4% including simple pavement, and the pavement rate of general national roads is about 99.5%. The spread of pavement can be expected to improve riding comfort, fuel consumption, and noise and vibration performance. Especially on paved roads, the friction coefficient is relatively high, and relatively high braking performance and obstacle avoidance performance are expected as a viewpoint of road safety. However, even on such pavement roads, these coefficients of friction have been shown to be significantly affected by surface conditions, such as wet, dry, snowy, and icy conditions, as well as running speed. It is also known that the friction coefficient of the road surface is greatly affected by the pavement material, pavement method, and usage conditions even if the road surface condition is the same. According to the literature [1], it is shown that the road surface friction coefficient significantly decreases according to the vehicle speed as shown in Figure 1. From Figure 1, it can be seen that the sliding friction coefficient in wet conditions depends on the speed, and in particular,

the reduction rate of the friction in the wet surface is extremely large. Furthermore, it was reported that the friction characteristics of the accident occurrence location were measured, and it was shown that the accident occurrence rate increases when the friction coefficient is 0.4 or less.

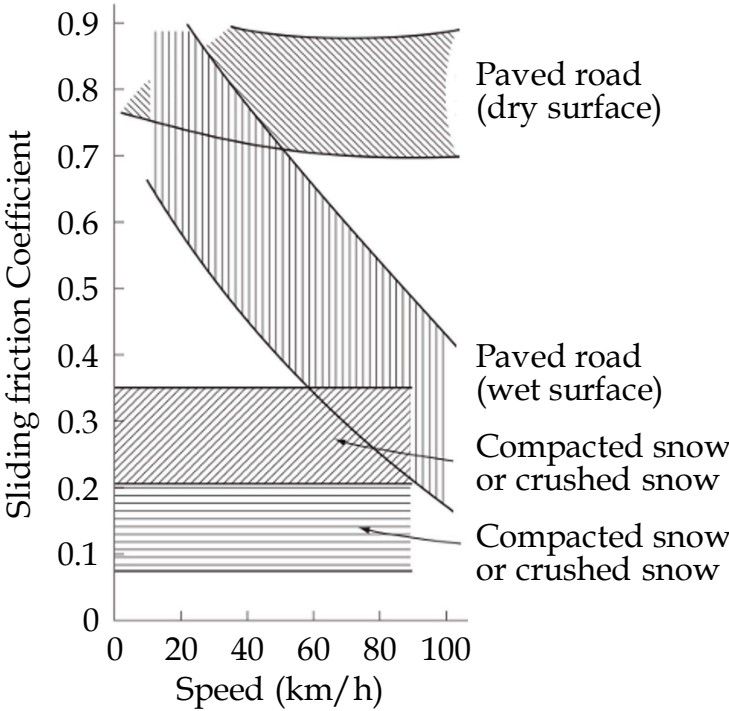

**Figure 1.** Sliding friction coefficient and road surface condition [1].

On actual roads, the coefficient of braking friction varies greatly depending on the state of wetness, running pattern of the vehicles, tire structure and surface material, and so on. According to researchers investigating pavement, pavement in recent years has taken safety into consideration, and the spread of permeable pavement, which is particularly widespread on highways in Japan, has led to a sharp decrease in the accident rate on highways. It is considered that this is largely due to the effect of improving visibility in rainy weather and suppressing a decrease in the friction coefficient on wet surfaces. In this way, the road surface friction characteristics are greatly related to the safety of the vehicle, and it seems that the safety has been secured considerably at present. In addition, these changes in friction characteristics are greatly affected by the dryness and wetness of the road surface, but a more dangerous condition is the effect of snowfall and freezing in winter, and in some cases, the friction coefficient decreases by about an order of magnitude. A systematic measurement method of the braking friction coefficient on an actual road and its creation in a database will be very important issues from the viewpoint of their contribution to traffic safety and construction of a new driving support system including automated driving. Therefore, it was pointed out that the importance of estimating the road surface in autonomous driving would increase in the future.

Therefore, in this study, we will provide information on how to consider the road friction coefficient in order to ensure the safety of the vehicle, especially the safety of the autopilot vehicle, which will become widespread in the future, and how to provide such information. Therefore, this study verifies the measurement algorithm based on the empirical results (using experimental results) for constructing a new road surface measurement system.

## 2. Characteristics of Road Friction

Normally, two main methods have been used to measure the road friction coefficient on the actual road surface: one is a stationary measuring device, such as a pendulum-type skid resistance tester (e.g., British Pendulum Skid Resistance Tester by Cooper Research Technology, Ripley, United Kingdom), DF tester (Dynamic Friction Tester by Nippo Sangyo, Kokubunji, Tokyo, Japan), and so on; and the other is a moving state measuring device, such as a bus-based or trailer-based road slip resistance measurement vehicle (e.g., Made-to-order products by Yachiyo Seisakusyo, Suginami, Tokyo, Japan), such as Grip Tester by Mastrad Limited, Douglas, United Kingdom, and so on [2,3]. The former is suitable mainly for measuring the dynamic friction coefficient at a local point, and the latter is suitable for measuring the friction characteristics of an actual tire while moving. The purpose of these devices is focused on measuring the maximum coefficient of friction on the road surface.

Important characteristics between the tire and road in the development of new vehicles include not only knowledge of the maximum coefficient of friction, but also knowledge of the value of the slip ratio at which the maximum coefficient of friction occurs, braking coefficient related to the braking effect, and rolling resistance related to fuel consumption. Figure 2 shows various road friction characteristics during braking measured at 65 km/h. In this study, many experiments were conducted to obtain many kinds of characteristics and shapes for the purpose of confirming the identification results of the *μ-s* characteristics on the various road surfaces. Figure 2 shows the results of measurements in dry and wet surface conditions using various road surfaces and tires, with the *Y* axis representing the friction coefficient and the *X* axis representing the slip ratio. The slip ratio shown here is shown as a percentage and is defined below as Equation (1):

$$s = \frac{v - \omega\, r}{v} \tag{1}$$

where *s*: slip ratio; *ω*: angular velocity of tire; *r*: tire radius; and *v*: vehicle speed.

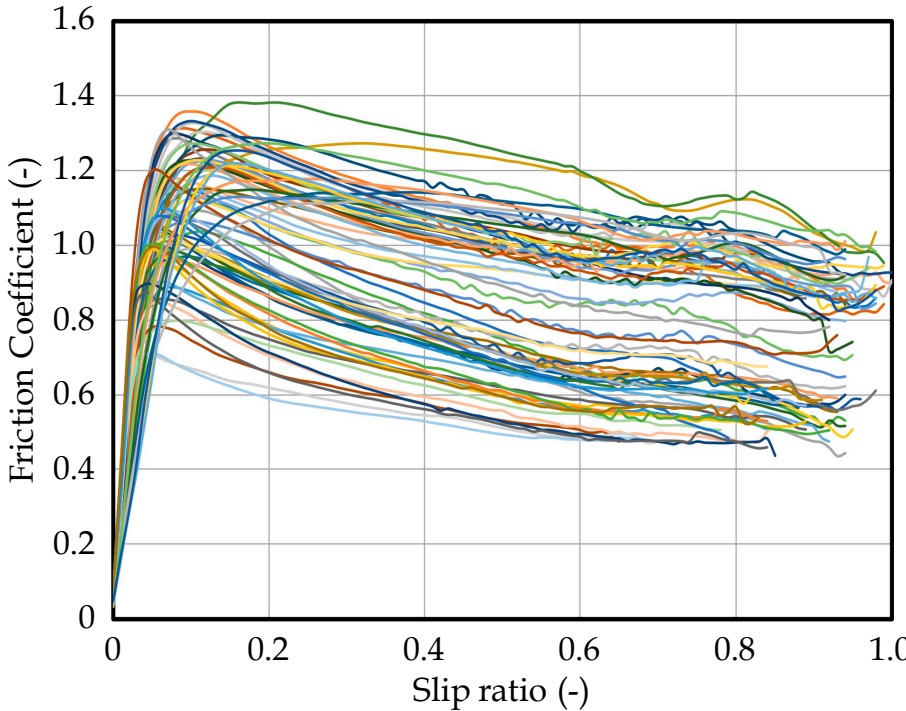

**Figure 2.** Characteristics of the road friction coefficient.

The sliding friction characteristics of the road with a wet surface shown in Figure 1 decreased sharply with respect to the speed, but from Figure 2, no significant difference in the characteristic shape in the region where the slip ratio was relatively small is shown. In

order to confirm the difference in these road surface conditions, the wet surface condition and the dry surface condition are classified, and the characteristic difference is shown in Figure 3. From Figure 3, it can be seen that the road friction characteristics on wet surfaces are significantly reduced compared to dry roads as a whole, and the characteristics shown in Figure 1.

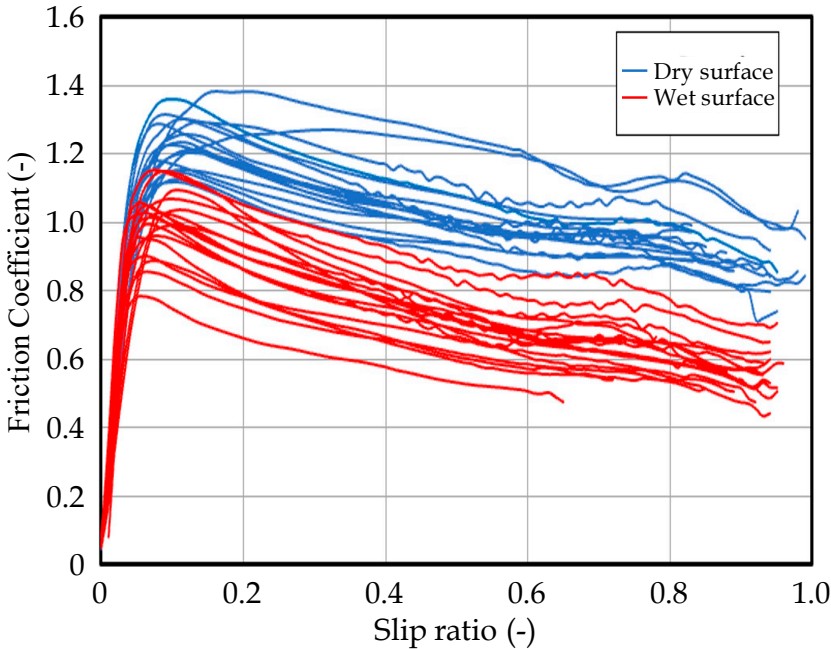

**Figure 3.** Difference in the friction coefficient depending on the road surface condition.

Figure 4 shows the characteristics of these properties with respect to the road friction and points that need to be examined. From the viewpoint of road safety, we need to evaluate the road friction using not only lock $\mu$, which has been measured as an evaluation value for various road surfaces in the field of road engineering, but also the peak $\mu$ and braking stiffness.

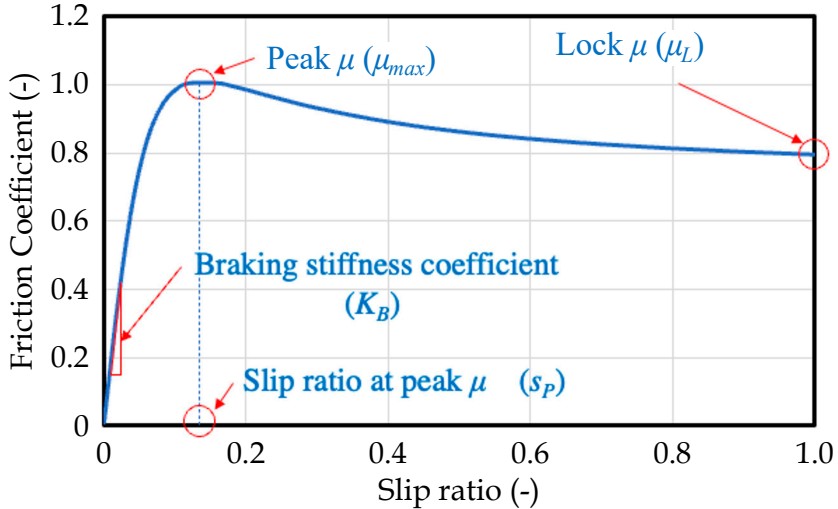

**Figure 4.** Important characteristics of road friction.

On wet roads, it seems that the peak $\mu$ is also reduced compared to dry surfaces, but this change is not big enough to affect road traffic safety. The reason for this is that an adhesion region and a sliding region are generated in the contact patch of the tire. The friction in the adhesive area is static and can be used up to the maximum static friction.

However, when the deformation of the tire exceeds the value allowed by the maximum friction, it becomes a slip region, and in this region, it changes to dynamic friction. The value of peak $\mu$ changes greatly depending on the difference between static friction and dynamic friction. Therefore, the decrease in peak $\mu$ on the wet road can be considered as a change of the ratio between the adhesive region and sliding region in the contact patch. In order to clarify the relationship around this point, we focused on the relationship between peak $\mu$ and lock $\mu$, which is shown in Figure 5.

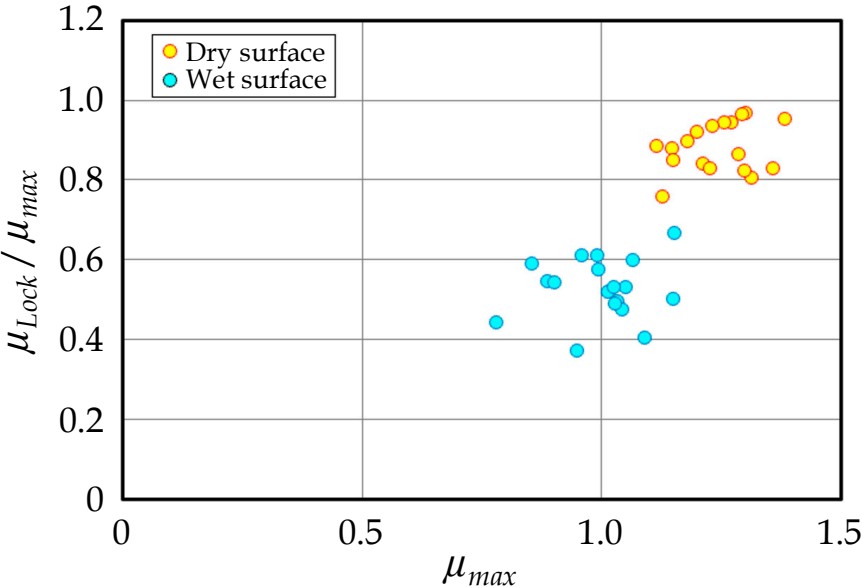

**Figure 5.** Ratio of lock $\mu$ to peak $\mu$.

In Figure 5, the value of peak $\mu$ is on the *X* axis, and the degree of the decrease in lock $\mu$ to the peak $\mu$ is shown with respect to this value. This data was measured at 65 km/h with a trailer-type tire testing machine. From Figure 5, the decrease in lock $\mu$ on the wet road surface is clearly shown, and it is considered that the effect of the decrease in the friction coefficient due to the velocity shown in Figure 1 is clearly shown. Currently, automobiles sold in Japan are obliged to install ABS, so when braking suddenly, such as emergency breaking, it will be used near the peak $\mu$. Therefore, in order to ensure vehicle safety, it is necessary to know not only the lock $\mu$ but also the peak $\mu$.

Another issue with road friction properties is how road position dependent these properties are. For this purpose, continuous friction characteristic measurement is required, but in the above-mentioned trailer type and other measuring instruments, the characteristic measurement is performed by changing the slip ratio during running, so this continuous data cannot be measured. The Grip Tester, as shown in Figure 6, can be used as a device that can measure continuously, but with this device, the slip ratio is fixed. In the $\mu$-$s$ characteristics shown in Figure 2, the slip ratios at peak $\mu$ appear to be around 10%; therefore, the measurement result in which the slip ratio on Grip Tester is fixed at 10% is shown in Figure 7. Here, the measurement result when traveling 250 m is shown. As it is clear from Figure 7, the road surface friction characteristic with respect to the position of the road surface changes greatly depending on the environmental change and the condition of the traveling vehicle. From these results, it can be seen that continuous measurement of road friction characteristics on ordinary road surfaces is important. However, another problem is that the peak $\mu$ changes depending on the road surface and tires. Figure 8 shows the relationship between the slip ratio $s$ at peak $\mu$ and the peak $\mu$ value using the $\mu$-$s$ characteristic shown in Figure 3.

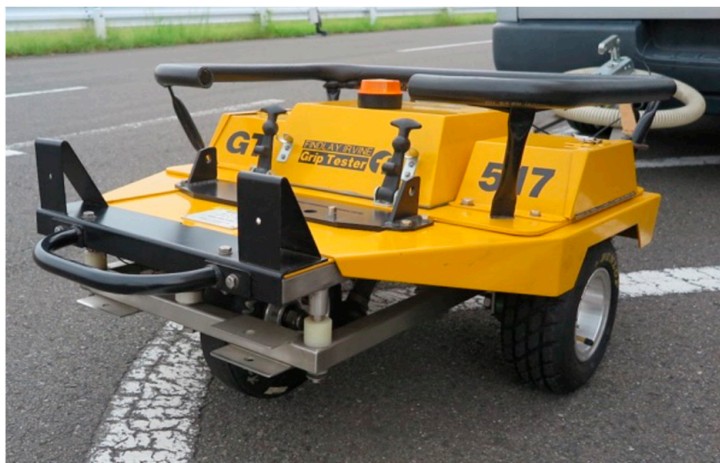

**Figure 6.** Grip Tester used in the experiment.

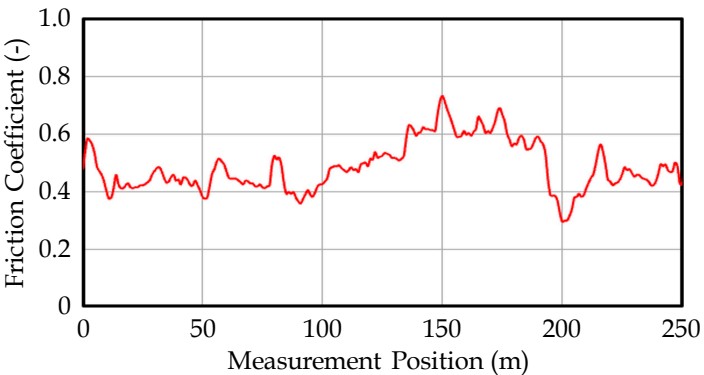

**Figure 7.** Measurement result for road friction with the Grip Tester.

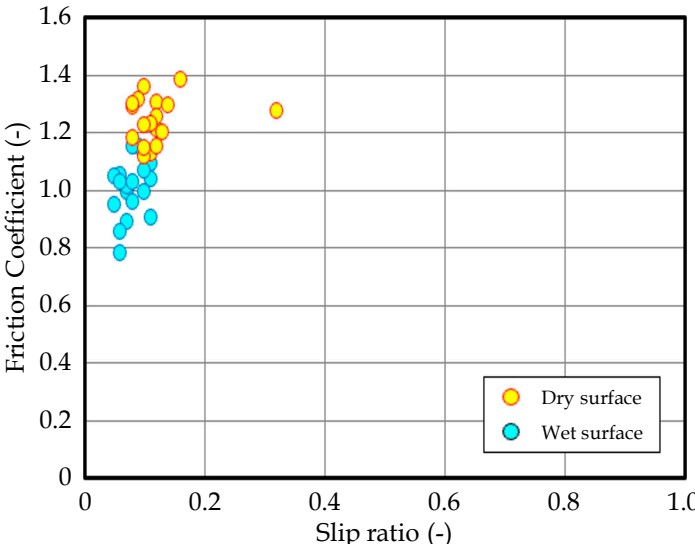

**Figure 8.** Slip ratio at peak $\mu$.

From Figure 8, it can be seen that the slip ratio at which $\mu$ peaks is generally concentrated at 20% or less, and these values change within this range. Therefore, it is important to continuously measure the $\mu$-$s$ characteristics. Therefore, Figure 9 shows an image diagram of this continuous $\mu$-$s$ characteristic and the data measured by the device currently used for these measurements.

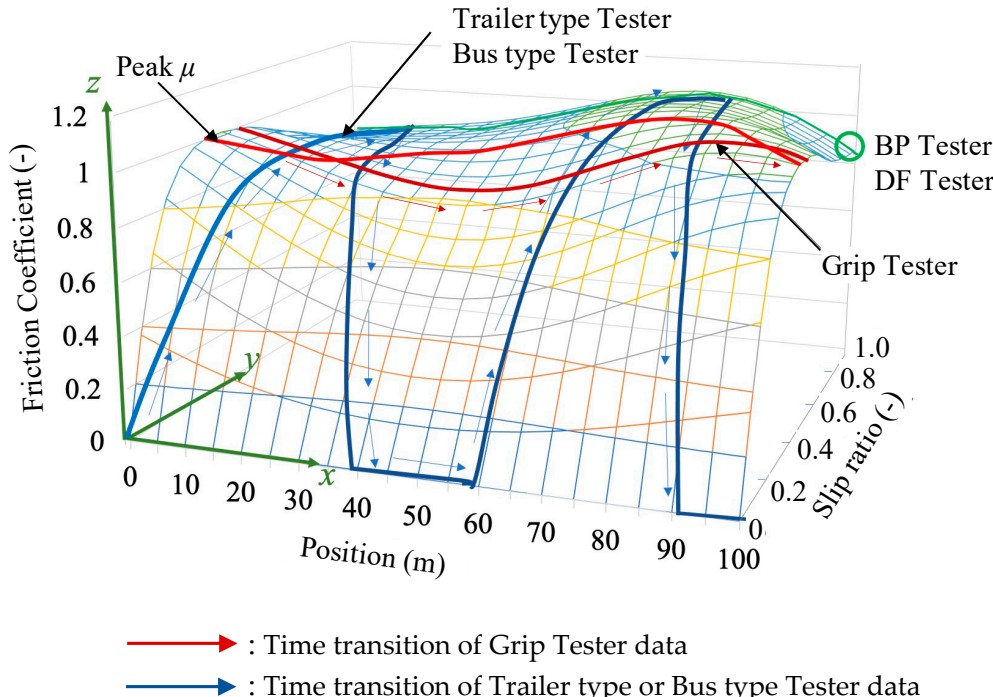

**Figure 9.** Image of $\mu$-*s* characteristics on actual roads and of the various measurement system data [4].

In Figure 9, the *y*-axis shows the slip ratio, the *z*-axis shows the friction coefficient of the road, and the *x*-axis shows the running position. Here, since the Grip Tester uses a sprocket and a chain to realize a substantially constant slip ratio with respect to the main tire, continuous data can be measured, but the slip ratio is constant. In general, trailer-type and bus-type testers that measure $\mu$-*s* characteristics apply braking force to the measured tires to change the slip ratio, and assuming that the friction during this length is constant, the measurement is performed with respect to the certain traveling distance as shown in Figure 9. Therefore, it is impossible to measure continuous characteristics of the travel distance. Furthermore, since the BP tester and the DF tester measure the lock $\mu$, they can be used as reference data for vehicle safety but not as effective information for the safety. Therefore, it is necessary to construct a method for measuring the $\mu$-*s* characteristics shown in Figure 9 with respect to the traveling direction of the road.

Finally, Table 1 summarizes the characteristics of devices that have generally been used to measure road surface friction characteristics in the past.

**Table 1.** Features of the measurement system.

| | Peak $\mu$ | Lock $\mu$ | $\mu$-*s* Characteristics | Continuity | Velocity Dependence |
|---|---|---|---|---|---|
| Trailer tester Bus tester | ○ | ○ | ○ | × | ○ |
| Grip tester | △ | × | × | ○ | ○ |
| BP tester | × | ○ | × | × | × |
| DF tester | × | ○ | × | × | ○ |

○: Measurable, △: Partially possible to measure, ×: Unmeasurable.

From this result, it can be seen that none of the measuring devices currently in use satisfy all the road friction information proposed in this study. Since there is no device for directly measuring the new road surface friction characteristics proposed by this paper, we propose a new measurement system for estimating these characteristics in the following sections.

### 3. Road Friction Estimation Method

In order to continuously measure the road surface friction characteristics, it is necessary to continuously measure the characteristics with a fixed slip ratio. Furthermore, it is necessary to confirm the method for estimating the overall $\mu$-$s$ characteristics from some combinations of the $\mu$ and $s$ set. In this study, the magic formula (MF) proposed by Prof. Pacejka in Delft University of Technology [5] is used to estimate the road friction characteristics. This equation is a function introduced from empirical research, and it is shown as a transcendental function. The original MF has been complicated and improved so that the detailed characteristics of the tire can be expressed, but this basic concept is used in this paper. The basis of this equation is a combination of the sin function and the arctangent function of Equation (2), which is expressed here using three parameters as follows:

$$\mu = a \sin\left\{b \ \tan^{-1}(c \ s)\right\} \tag{2}$$

where $a$, $b$, and $c$ are parameters for determining the characteristic shape, and $s$ is the slip ratio. Since it is necessary to identify the entire property with this MF, we need to identify the parameters $a$, $b$, $c$ using at least 3 sets of $\mu$-$s$ values. To clarify the relationship between these characteristics and each parameter, the MF is differentiated by the slip ratio $s$ to obtain the following Equation (3):

$$\frac{d\mu}{ds} = \frac{a \ b \ c \cos\left\{b \ \tan^{-1}(c \ s)\right\}}{c^2 \ s^2 + 1} \tag{3}$$

Here, since the differentiation of MF at $s = 0$ represents $K_B$, the following Equation (4) is introduced:

$$K_B = \left.\frac{\partial \mu(s)}{\partial s}\right|_{s=0} = a \ b \ c \tag{4}$$

Next, at the point where Equation (3) becomes 0, the value at which this value peaks is peak $\mu$ ($\mu$-max), so the slip ratio at which peak $\mu$ occurs using this is shown by Equation (5):

$$s_P = \frac{\tan\left(\frac{\pi}{2b}\right)}{c} \tag{5}$$

Using Equation (5), the peak $\mu$ ($\mu$-max) is expressed by the following Equation (6):

$$\mu_{\text{max}} = a \sin\left[b \ \tan^{-1}\left\{\tan\left(\frac{\pi}{2b}\right)\right\}\right] \tag{6}$$

Further, lock $\mu$ is obtained by substituting $s = 1$ for Equation (2), and is shown in the following Equation (7):

$$\mu_L = a \sin\left(b \ \tan^{-1} c\right) \tag{7}$$

In order to express each characteristic shown here, it is necessary to identify the parameters $a$, $b$, and $c$ in the MF shown in Equation (2), using the experimental data. We use the experimental results of three sets of $\mu$ and $s$. This analysis is carried out by iteration using the steepest descent method programed by MATLAB. As an example, Figure 10 shows a comparison between the result identified using MF and the experimental result. From this result, it can be seen that although the value of lock $\mu$ is slightly different, the $\mu$-$s$ characteristics can be generally identified by using MF. The parameters of this identification result are as follows:

$a = 0.8232$, $b = 1.6450$, $c = 21.1680$.

Therefore, the results of determining these three parameters $a$, $b$, $c$ based on the obtained experimental data are shown in Figure 11. Since the parameter $a$ represents the value of the peak $\mu$, it is compared with this value. Here, since the identification is performed using the steepest descent method, the relationship with other parameters is also examined, but from the relationship in Figure 11a, the correlation coefficient between

$\mu$-max and the parameter $a$ is about 0.994. It shows a very high correlation. The relationship between the remaining parameters $b$ and $c$ is shown in Figure 11b, and although there are large variations due to differences in shape, etc., the range that these parameters can take is clarified. MF is a transcendental function, and if it is calculated using the data of the three points in this paper, innumerable solutions that completely match these will occur, but it is impossible to describe actual $\mu$-$s$ characteristics using such a solution. So, the convergence calculation is performed by specifying the range that each parameter can take with reference to these values. Therefore, identification is performed by setting restrictions on each coefficient based on the shape of the actual $\mu$-$s$ characteristic.

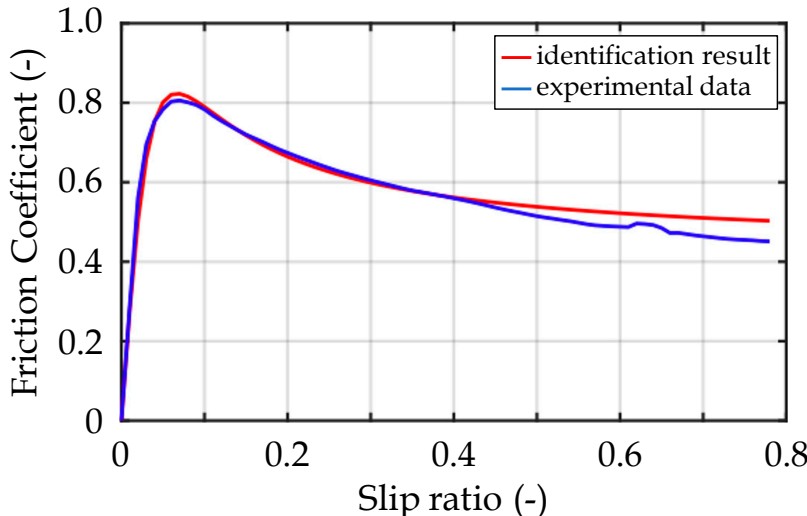

**Figure 10.** Comparison between the experiment and identification results [4].

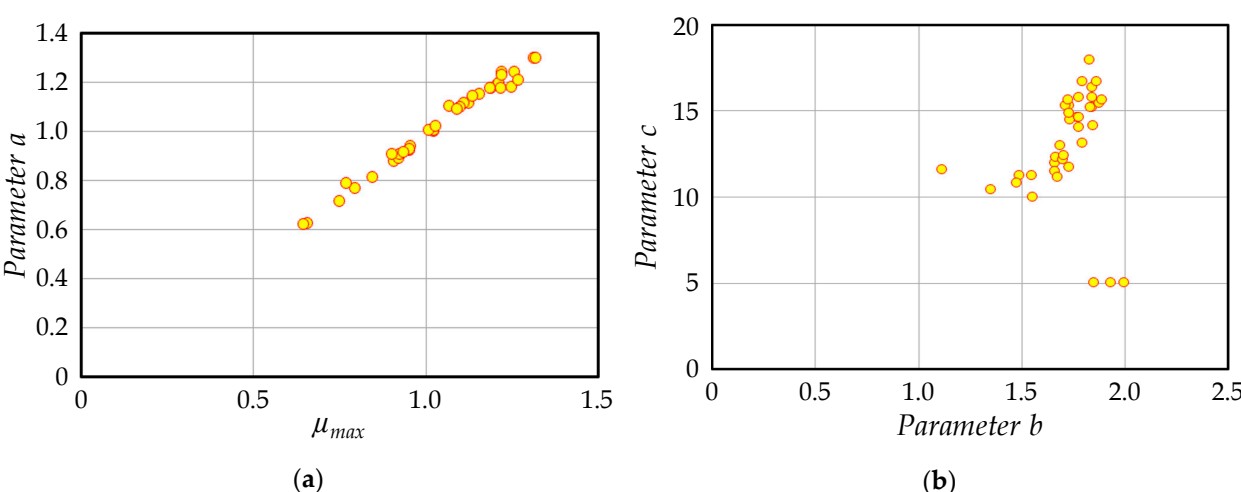

(**a**)　　　　　　　　　　　　　　　　　　　(**b**)

**Figure 11.** MF parameters identified from experimental data. (**a**) Relation between $\mu_{max}$ and parameter $a$. (**b**) Relation between parameter $b$ and $c$.

## 4. Identification Result of Road Friction Characteristics

In Figure 8, the slip ratio $s_P$ at which the peak $\mu$ occurs is concentrated at about 5% to 10% in the wet surface conditions and around 8% to 18% in the dry surface conditions. Since most vehicles in Japan are equipped with an ABS system, the operating range when emergency braking is required is mainly near this peak $\mu$. In order to improve the estimation accuracy of peak $\mu$ as much as possible, it is necessary to concentrate around the slip ratio $s_P$ for the identification. Therefore, the slip ratios for measurement are set to 3, 10, and 17% for the equipment designed in this paper. Therefore, the results of estimating the $\mu$-$s$ characteristics using the characteristics of these three points are shown for the

characteristics of the three different characteristics as shown in Figure 12. Figure 12 shows the result of the identification in consideration of the above-mentioned parameter setting region, and the three points in Figure 12 are the values used for identification, and the blue line shows the experimental results and the red line shows the identification results using MF. It can be seen that $K_B$, $\mu$-max, $s_P$, etc. are well represented. Therefore, regarding the experimental results conducted in the past, the comparison between the results obtained by identification and the experimental results is shown in Figure 13. These characteristics of road friction are very important factors for traffic safety, such as vehicle behavior and motion control, and so on. Focusing on these identification results, it can be seen that the correlation with the experimental results is very high (each correlation coefficient is more than 0.97) and can be an important index for traffic safety. Therefore, next, we examine a system that simultaneously measures these three points.

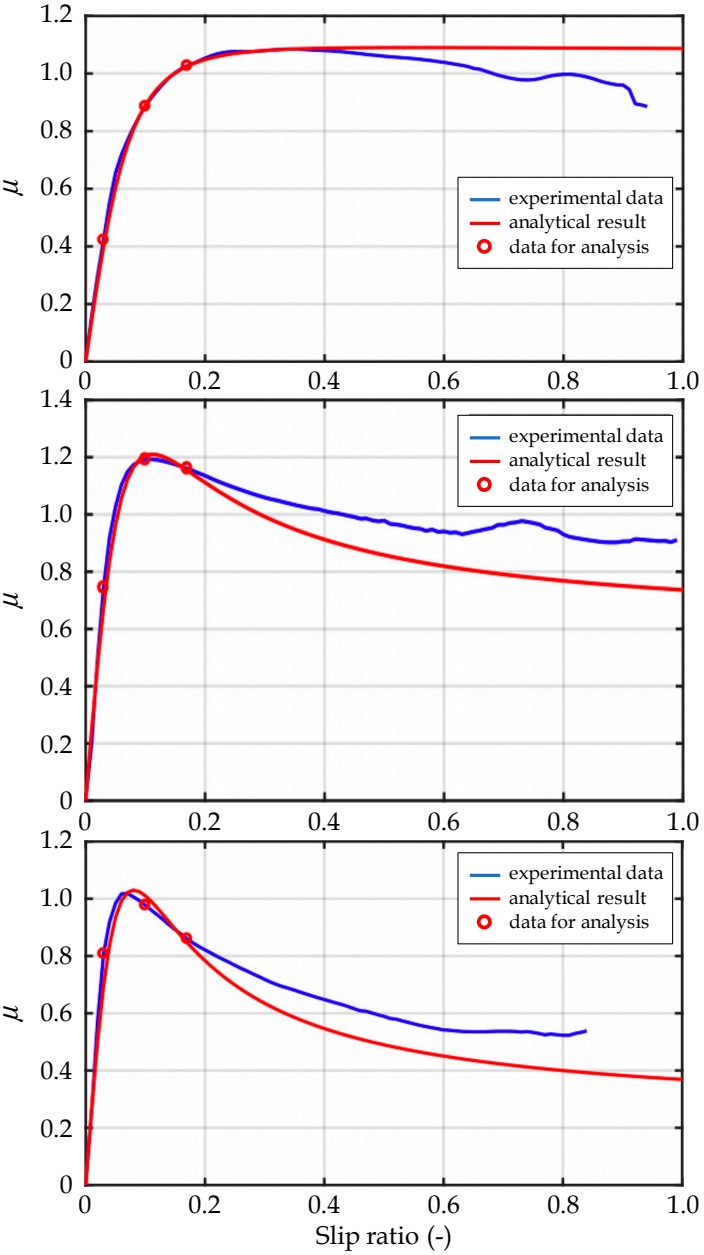

**Figure 12.** Identification results of $\mu$-$s$ characteristics by three different points ($s$ = 3, 10, 17%).

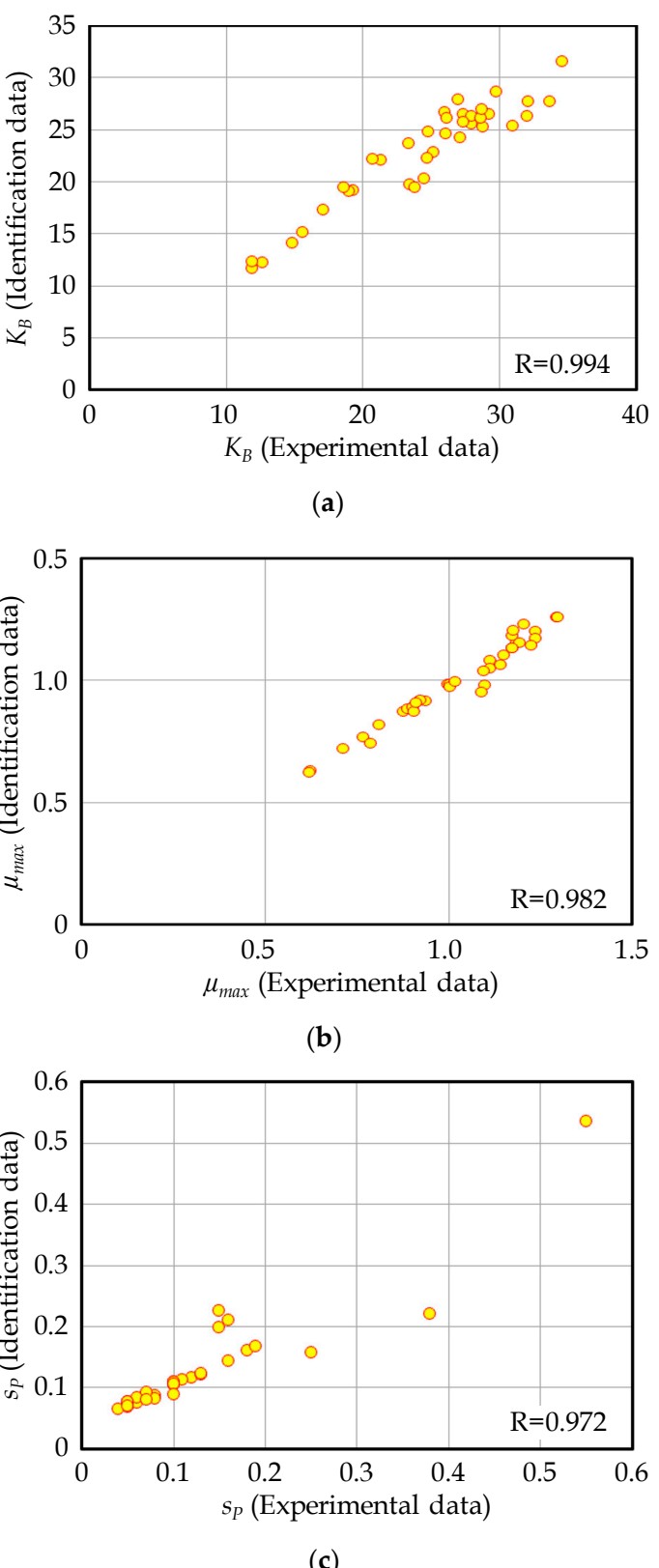

**Figure 13.** Identification results. (**a**) Identification result for $K_B$. (**b**) Identification result for $\mu_{max}$. (**c**) Identification result for $s_P$.

## 5. Consideration of the Measurement Device

It is important to consider a mechanical connection to rotate the tire with a constant slip ratio. As a simple mechanism, it can be realized by arranging sprockets or a toothed pulley with different numbers of teeth on the two tires shown in Figure 14, and connecting them with a chain or a timing belt. In Figure 14, tire 1 is the main tire (driving tire) and tire 2 is the measurement tire. What may be a problem here is that the main tire may be greatly affected by the measurement tire and may not be able to rotate sufficiently. Since these tires are attached to a trailer or a vehicle, they have the same speed, but it is necessary to calculate the slip ratio of each tire in this case. Here, the same tires are used (assumption $R_1 = R_2$), the sprocket radii ($r_1$, $r_2$) are changed, and the slip ratios of each tire are calculated as $s_1$ and $s_2$ from the vehicle speed $v$ and the respective speeds $\omega_1$ and $\omega_2$. Here, it is assumed that the rolling resistance is small, and that the $\mu$-$s$ characteristics of each tire are the same. In this case, the slip ratio of the measured tire to the main tire is represented below as Equation (8) using the mechanical connection condition:

$$s_2 = 1 - \frac{r_1}{r_2}(1 - s_1) \tag{8}$$

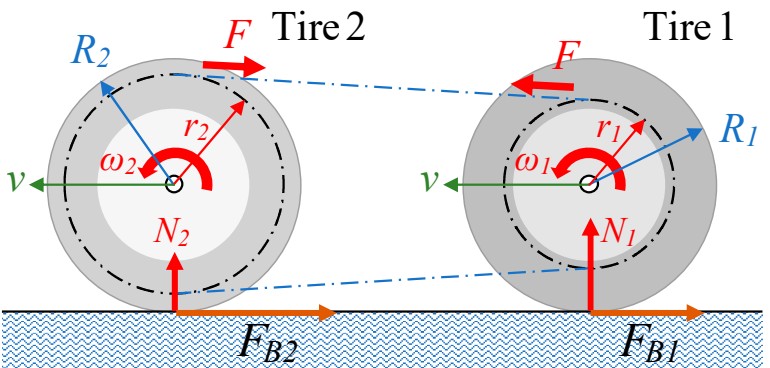

**Figure 14.** Mechanical properties between two tires.

In addition, the following Equation (9) can be obtained from the relationship between the torque balance of individual tires and the tension of the chain:

$$F_{B1} = -\frac{r_1}{r_2}F_{B2} = -\frac{r_1}{r_2}\left[a \sin\left\{b \tan^{-1}(c\,s_2)\right\}\right] N_2 \tag{9}$$

Tire 1 becomes a drive tire, and the slip ratio can be kept small due to the sprocket ratio. Therefore, under the condition ($s_1 << 1$), the longitudinal force of tire 1 can be approximated as follows:

$$F_{B1} \cong a\,b\,c\,s_1 N_1 \tag{10}$$

Transforming Equation (10) above, the slip ratio of the main tire is described as follows as Equation (11):

$$s_1 = -\frac{1}{b\,c}\frac{r_1}{r_2}\frac{N_2}{N_1}\sin\left\{b \tan^{-1}(c\,s_2)\right\} \tag{11}$$

Since the measurement tires investigated in this study have the configuration shown in Figure 15, it is necessary to extend these equations. It is assumed that each load $N_s$ of the sensing tire is the same:

$$s_1 = -\frac{r_1}{b\,c}\frac{N_s}{N_1}\left[\frac{\sin\left\{b \tan^{-1}(c\,s_2)\right\}}{r_2} + \frac{\sin\left\{b \tan^{-1}(c\,s_3)\right\}}{r_3} + \frac{\sin\left\{b \tan^{-1}(c\,s_4)\right\}}{r_4}\right] \tag{12}$$

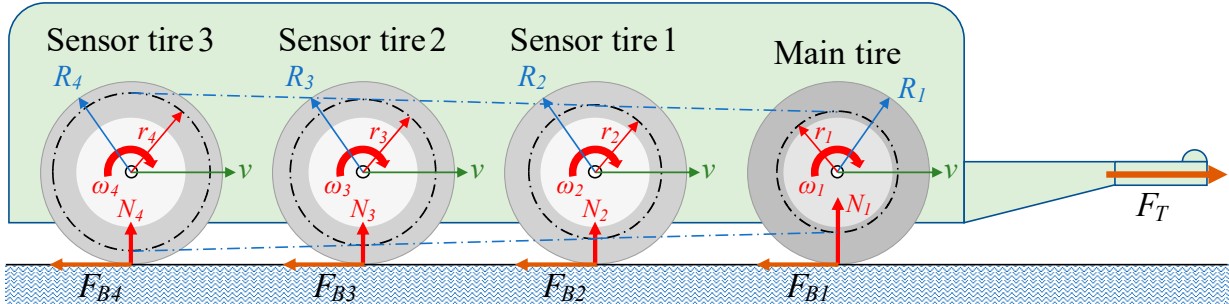

**Figure 15.** Trailer-type measurement system.

As an example, Figure 16 shows the results of a calculation with tire loads of 500 N, $b = 1.7$, and $c = 12$. In addition, since braking force is always applied to these measured tires, it is necessary to calculate the traction force *FT*, but when calculating the longitudinal forces of four tires from Equation (12), it is approximately 1.2 kN. From this value, it is considered that the traction force is not a value that hinders normal running, and that this system functions sufficiently.

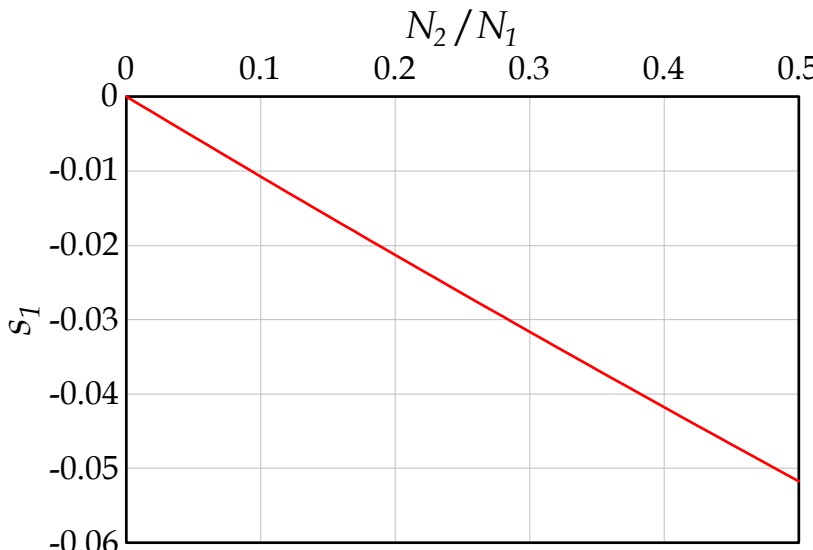

**Figure 16.** Slip ratio of tire 1.

## 6. Flow of Supply for Research Results

The outline of the identification method explained in this paper and the method of providing the results to society are as follows:

1.  The tire characteristic measurement system discussed in Chapter 5 will be constructed, and the continuous characteristics of three sets of $\mu$ and $s$ will be measured using this system.
2.  Using these data sets, the parameters *a*, *b*, and *c* at each measuring point are identified using the steepest descent method according to the recording timing. The basic idea of this is shown in Section 3. In this convergence calculation, the zones *a*, *b*, and *c* shown in Figure 11 are used, but in consideration of expandability, the convergence calculation is performed in the region $\pm 1.5$ times these values.
3.  Using each of the obtained parameters, the braking stiffness, the peak $\mu$ values, its slip ratios, and the lock $\mu$ at each point are calculated. The basic image of the database to be constructed is shown in Figure 9.
4.  A database of these will be created for each road, and will be provided to autopilot vehicles or vehicles using ADAS.

### 7. Conclusions Remarks

This study examined the construction of a device for continuously measuring the friction characteristics of actual roads, which are greatly related to road safety. As a result of the analysis, the following conclusions were obtained:

1.  The equipment that measures road friction characteristics were examined, especially continuously measurement for $\mu$-$s$ characteristics, and it was shown that sufficient information cannot be obtained with the current equipment.
2.  It was shown that identification using three sets of continuous measurement results of $\mu$-$s$ characteristics using MF can provide identification results that can sufficiently contribute to traffic safety.
3.  Finally, the outline of the trailer-type measuring device using the method was shown.

We are currently constructing this measuring device, and we plan to show the results in future research. As a future task, it is necessary to confirm the accuracy of the data obtained from the constructed measurement vehicle. Furthermore, it is necessary to construct an estimation system of forward road friction characteristics by using environmental information measured at the same time as the road surface friction database to be constructed.

**Author Contributions:** Conceptualization, T.H. and I.K.; Data curation, I.K. and G.M.; Formal analysis, I.K. and Y.K.; Funding acquisition, T.H., I.K. and M.A.; Investigation, T.H. and T.K.; Methodology, I.K. and Y.K.; Project administration, T.H.; Resources, T.K.; Validation, Y.K., M.A. and G.M.; Writing—original draft, I.K.; Writing—review & editing, T.H. All authors have read and agreed to the published version of the manuscript.

**Funding:** This research received no external funding.

**Institutional Review Board Statement:** Not applicable.

**Informed Consent Statement:** Not applicable.

**Acknowledgments:** A part of this research was carried out as a research project of East Nippon Expressway Company Limited (NEXCO East Japan). We would like to thank all the people involved.

**Conflicts of Interest:** The authors declare no conflict of interest.

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
