# Peer review of "Study on the Road Friction Database for Automated Driving: Fundamental Consideration of the Measuring Device for the Road Friction Database"

_applsci, doi:10.3390/app12010018_

Round 1

Reviewer 1 Report

Dear authors,

Thank you for submitting your work to the MDPI journal Applied Sciences. 

Your submitted paper entitled "Study on Road Friction Database for Automated Driving" explores a method for continuously measurement for the road friction characteristics such as μ-s characteristics. The idea of this paper is valuable and the paper may contribute to the existing body of research.

However, thoroughly revise your paper prior to publication. Currently, it cannot be published. Despite the minor issues, address and thorouhgly correct all major issues and re-submit your paper.

Thank you!

Minor issus: 
Abstract
1) improve the English writing: "In an automated driving such levels, the"
2) at least one sentence on your conclusions should be included

Major issues:
(1) In the methodology section it is not clear whether this study is a simulation study (as it can be assumed) or an empirical study. 
(2) Missing: State the source of the data. Where were the data simulated (or collected)? 
(3) The study "design" section is totally missing. Include it, even if it might be a simulation study. Then describe your design to enable other researchers to potentially replicate your analyses.
(4) State which software packages was used for your analyses.
(5) Share your (simulated) data set transparently with the scientific community either on the MDPI platform, or on OSF. - So far, you compute numerous values in the 16 (!) Figures, but no one can prove and re-evaluate them so far.
(6) Share your computation code too and comment on it that the scientific community can test and re-use it, in case it works well.
(7) Include the new section "Discussion"
(8) Include the new section "Limitations and Future Research" - List all your identified limitations and reflect on it/provide possible solutions in future research before you state your conclusions

Reviewer 2 Report

In my opinion, the article has got potential. However, the article looks inconsistent, and not all parts, results, and interpretations are completely clear and understandable to the reader.

For the background and references, they need to be edited, I suggest adding the following references not yet mentioned in the article but which are highly relevant to it:

Jorge Villagra, Brigitte d’Andréa-Novel, Michel Fliess, Hugues Mounier. A diagnosis-based approach for tire-road forces and maximum friction estimation. Control Engineering Practice, Elsevier, 2011, 19 (2), pp.174-184. 10.1016/j.conengprac.2010.11.005. inria-00533586

Juan A. Cabrera, Juan J. Castillo, Javier Pérez, Juan M. Velasco, Antonio J. Guerra and Pedro Hernández. A Procedure for Determining Tire-Road Friction Characteristics Using a Modification of the Magic Formula Based on Experimental Results. Sensors 2018, 18, 896; doi:10.3390/s18030896

or:

Sara Yeni González Endrinal. Modelling the wheels of the Robot MAX2D and surfacing. Master thesis. Hochschule Ravensburg-Weingarten. 2010.

I have a few comments and questions:

Line 83 – How, or with what measurement device was measured coefficient of friction when even on wet surfaces are obtained values higher than 1,0? According to my opinion, it is not much typical.

Line 85 -86 – “... with the longitudinal axis representing the friction coefficient and the lateral axis representing the slip ratio.“ – vice versa, I recommend using description as X- and Y-axis.

Line 102 – 103 – „On wet roads, it seems that the peak μ is also reduced, but not so big change to affect road traffic safety.“ – Comparing to what? Lock? Dry surface? Or at different speeds? It is not clear.

Line 103 – 105 – „The reason for this is that an adhesion region and a sliding region are generated in the contact patch of the tire, and the reason why friction coefficients in the sliding region is reduced at wet surfaces.“ – The sentence is also not clear. Under wet conditions, the adhesion component of the total friction is drastically reduced, whereas the hysteresis loss remains largely unaffected.

Line 112 – „...the value of peak μ is on the lateral axis, ...“ – see comment above, X-axis.

Line 114 – 116 – I wouldn't agree with that. Yes, the decrease in friction coefficient due to the velocity shown in Figure 1 is clearly shown and it is a widely known fact, however, there are also differences between different surfaces depending on a combination of factors like good and bad macrotexture and microtexture what can influence the speed of change. But, according to Figure 3, the degree of decrease is almost the same at all surfaces. The slope of the curve is almost the same, the absolute difference between μ -peak values for different surfaces is very similar (about 0.3-0.4, at least what I can read from these confusing charts - I recommend choosing only a few lines for typical pavements and changing colors like more shades of red and blue) and the same for and μ -lock values. But more importantly, the difference makes the type of surface and values measured on them, and what we take to the proportion 0.2/0.6; 0.4/0.8; or 0.8/1.0. This makes the difference, but wouldn't be better for the evaluation of the degree of decrease to compute: (μ -peak - μ -lock) / (s-peak – s-lock)?

Line 118 – 119 – Therefore, in order to ensure a vehicle safety, it is necessary to know not only the lock μ but also the peak μ.“ – in most countries is usually measured just the peak μ.

Line 193 – 199 – Please, describe experimental measurements for identifying parameters a, b, c. Please, clarify what does it mean, three sets of μ and s? How many different pavements were tested, with what levels of micro and macrotexture, at what speed, with what tire, etc.? Please, describe the procedure how the parameters were determined using experimental results?

Line 209 – 210 – „...the range that these parameters can take is clarified.“ – The range for parameter c is, according to results and Figure 11b from 5 to 18, but at line 199 is the value of parameter c =  21.1680, why?

Line 234 – „... It can be seen that KB, μ-max, sP, etc. are well represented.“ – In Figure 12 is well-represented at most KB, but μ-max and sP, are obviously not.

Round 2

Reviewer 1 Report

Dear authors,

the revised version is better. 

However, in the present form it cannot be published.

The following issues have not been addressed:

1) State which software packages was used for your analyses + "include the exact version number". 
2) Share your (simulated) data set transparently with the scientific community on OSF: https://osf.io/ and include the link in your paper - So far, you compute numerous values in the 16 (!) Figures, but no one can prove and re-evaluate them so far.
3) Share your computation code too on OSF and comment on it that the scientific community can test and re-use it. Provide more transparent insights.
4) Include the new section "Limitations and Future Research" and professionally list all your identified limitations and reflect on it/provide possible solutions for future research before you state your conclusions.

Thank you!
